# Tumor Location and a Tumor Volume over 2.8 cc Predict the Prognosis for Japanese Localized Prostate Cancer

**DOI:** 10.3390/cancers14235823

**Published:** 2022-11-25

**Authors:** Haruki Baba, Shinichi Sakamoto, Xue Zhao, Yasutaka Yamada, Junryo Rii, Ayumi Fujimoto, Manato Kanesaka, Nobuyoshi Takeuchi, Tomokazu Sazuka, Yusuke Imamura, Koichiro Akakura, Tomohiko Ichikawa

**Affiliations:** 1Department of Urology, Chiba University Graduate School of Medicine, Chiba 260-8670, Japan; 2Department of Urology, Japan Community Health-Care Organization Tokyo Shinjuku Medical Center, Tokyo 162-8543, Japan

**Keywords:** tumor volume, tumor location, prostate cancer, biochemical recurrence, prognostic factor

## Abstract

**Simple Summary:**

About 40% of men with localized prostate cancer experience biochemical recurrence after radical prostatectomy. The early detection of disease progression is important for optimal post-operative treatment and follow-up. Our study reviewed 557 patients with prostate cancer who underwent radical prostatectomy and found that a tumor volume over 2.8 cc was a novel independent predictive factor for biochemical recurrence. We further established a novel risk assessment model based on tumor volume and location (posterior and peripheral zone). We confirmed that the risk model could stratify patients’ prognoses. In addition to the previously reported biomarkers, these novel factors obtained from the surgical specimen may provide better prognostic information in patients with prostate cancer.

**Abstract:**

(1) Objective: Our study investigated the prognostic value of tumor volume and location in prostate cancer patients who received radical prostatectomy (RP). (2) Methods: The prognostic significance of tumor volume and location, together with other clinical factors, was studied using 557 patients who received RP. (3) Results: The receiver operating characteristic (ROC) curve identified the optimal cutoff value of tumor volume as 2.8 cc for predicting biochemical recurrence (BCR). Cox regression analysis revealed that a tumor in the posterior area (*p* = 0.031), peripheral zone (*p* = 0.0472), and tumor volume ≥ 2.8 cc (*p* < 0.0001) were predictive factors in univariate analysis. After multivariate analysis, tumor volume ≥ 2.8 cc (*p* = 0.0225) was an independent predictive factor for BCR. Among them, a novel risk model was established using tumor volume and location in the posterior area and peripheral zone. The progression-free survival (PFS) of patients who met the three criteria (unfavorable group) was significantly worse than other groups (*p* ≤ 0.001). Furthermore, multivariate analysis showed that the unfavorable risk was an independent prognostic factor for BCR. The prognostic significance of our risk model was observed in low- to intermediate-risk patients, although it was not observed in high-risk patients. (4) Conclusion: Tumor volume (≥2.8 cc) and localization (posterior/peripheral zone) may be a novel prognostic factor in patients undergoing RP.

## 1. Introduction

Prostate cancer (Pca) is the most common malignant tumor in men. About 2.6 million cases are newly diagnosed and 34,500 deaths of Pca are estimated per year in the United States [1]. Radical prostatectomy (RP) for the treatment of prostate cancer has made remarkable progress since it widely emerged around 1900. At present, RP is still the standard treatment option for localized Pca [2]. However, the frequency of biochemical recurrence (BCR) has been reported to be about 40% within 10 years after RP [3]. Once BCR occurs, about 3.5% of patients will inevitably develop resistance to androgen deprivation therapy, also known as castration-resistant prostate cancer (CRPC) [4]. CRPC has been reported to cause death within 2 to 4 years [5]. Therefore, BCR is the major clinical issue to be detected and addressed in patients who received RP.

A lot of clinical studies have evaluated predictive factors and/or risk models for BCR after RP. Serum prostate-specific antigen (PSA) is the mainstay to detect the BCR of patients after surgery [6], and it has been recommended to keep close monitoring until PSA reaches 0.2 ng/mL [7]. In addition to PSA kinetics, Gleason score, PSA density, pathological and clinical stages, surgical margin, and other clinical factors have been studied for their prognostic significance, however, these factors could not predict BCR independently [8]. To better distinguish the recurrence risk and evaluate the prognosis after RP, more innovative predictors or models are unmet clinical needs. The individualized management after treatment requires effective recurrence risk prediction to implement timely intervention and avoid overtreatment. Previous studies showed that the tumor volume was related to the clinical manifestations of prostate cancer [9]. A tumor with a volume of less than 0.5 cc is considered as insignificant prostate cancer, and aggressive treatment may not be needed [10,11]. Recently, several studies proposed the novel definition of insignificant prostate cancer as a tumor volume of less than 2.5 cc [11,12,13,14,15,16,17], or less than 2.0 cc [18]. However, it was found that the BCR risk increased with tumor volume over 2.49 cc, indicating that the tumor volume was deeply involved in the progression of Pca [19]. Furthermore, little is known about the relationship between different prostate areas and tumor volumes, and their impact on BCR. Herein, we examined the prognostic role of tumor volume and location in patients with localized Pca for a better treatment strategy and postoperative follow-up.

## 2. Methods

### 2.1. Study Design and Setting

Clinical data of 557 patients who received RP at Chiba University Hospital and affiliated hospitals between 2006 and 2020 were retrospectively reviewed. The study was approved by the clinical review committee of our institution (#1768) and the written informed consent of all patients participating in the study was obtained. All participants or designated agents accepted a standardized data collection protocol, including personal postoperative follow-up information and medical record. The study is in accordance with the Japanese ethical document.

### 2.2. Patients

The inclusion criteria were RP for biopsy-proven prostate cancer performed at Chiba University Hospital and affiliated hospitals; whole-mount step-section pathologic maps available for tumor volume-calculation and localization. The exclusion criteria were neoadjuvant hormone therapy; radiation therapy; poor pathologic map quality; short follow-up term (<12 months).

### 2.3. Variables

Baseline clinical data included age, BMI, serum PSA, PSA F/T ratio, serum testosterone, biopsy positive rate, Gleason score (GS), clinical TNM staging, surgical prostate specimen, tumor volume, tumor location, surgical resect margin, and pathological TNM staging. Each patient came to our institution every 3 months after RP and had blood samples taken for PSA measurement until the occurrence of BCR or death was confirmed.

After RP, an elevated serum PSA level (>0.2 ng/mL) was defined as BCR [6].

### 2.4. Tumor Volume and Location Estimation Method

#### 2.4.1. Measurement of Tumor Volume

The prostatectomy specimens were step-sectioned transversely at 5-mm intervals. All the specimens were mounted on slides. Tumor volume was calculated by scanning the sliced specimen, and the area of the tumor was analyzed using ImageJ software. Total tumor volume = tumor area × thickness of specimen × 1.2 (correction for shrinkage).

#### 2.4.2. Tumor Localization

All specimens were serially sectioned from the tip to the base at 5 mm intervals, and the bladder neck and vertex edges were submitted as vertical sections. According to the anatomical structure, the specimen was divided into the following regions: the peripheral zone (PZ), the transition zone (TZ), and the central zone (CZ). The region within 1.0 cm or 1.5 cm from the tip of the prostate was identified as the Apex region. The prostatic urethra is an anatomic marker for a tumor to be classified as anterior or posterior (Figure 1). If a tumor showed a slight extension to another site, >80% volume in the main area was the criterion for defining the origin of the tumor in this area. Each RP sample was reviewed by two pathologists.

### 2.5. Statistical Methods

JMP Pro (Version 16.0; SAS Institute Inc., Cary, NC, USA) was used for statistical analysis. Univariate cox proportional hazards model analysis was performed on the baseline data classified by the median value of the outcome measurement to determine the predictive factors of the BCR. The significant variables (*p* < 0.05) were further analyzed by multivariable cox proportional hazards model regression. The optimal cutoff value of tumor volume was obtained by calculating Area Under the Curve (AUC) from the Receiver Operating Characteristic (ROC) curve analysis. To evaluate the interaction between tumor volume and location, 3 risk factors related to volume and location obtained from univariate and multivariate cox regression analysis were combined into a risk classification model. This model was grouped according to the number of risk factors displayed: favorable; 0 risk factor, moderate; 1 or 2 risk factors, unfavorable; all 3 risk factors. Kaplan–Meier method was used to evaluate progression-free survival (PFS). Statistical significance was set at *p* < 0.05.

## 3. Results

### 3.1. Participants

In total, 557 patients were enrolled in the study. Follow-up terms ranged from 12 to 161.5 months, with a median follow-up time of 45.3 months. As of the end of the study, 66 (11.8%) patients had BCR, and 9 (1.6%) patients died (not due to prostate cancer). The median age of all patients was 67 years old. The median preoperative PSA level was 7.71 ng/mL. The biopsy GS was 7 or less in 79.7%, 8 in 8.6%, and 9 or more in 11%. Overall, 64.8% of patients were pathological TNM stage 2c or above, and 1.4% were positive for lymph node metastasis. According to the risk grouping of Pca by the American Cancer Society (ACS), 77 (13.8%) patients were classified into the low-risk group, 279 (50.1%) were classified into the intermediate-risk group, and 201 (36.1%) were classified into the high-risk group. The median tumor volume was 2.12 cc. Seminal vesicle invasion was observed in 8.6%, the extracapsular invasion was seen in 24.8%, and 30.3% had positive margins. The tumor distributions were in the apex area (63.7%), middle area (63.4%), and bladder neck (21.4%). Regarding the anterior or posterior area of the prostate, 48.1% of the tumors were in the anterior, and 52.4% were in the posterior. Overall, 67.1% were located in the PZ and 37.3% were in the TZ (Table 1).

### 3.2. Predictive Factors for Progression-Free Survival (PFS)

The ROC curve was used to calculate the relationship between BCR and tumor volume, and the optimal cutoff value was identified as 2.8 cc (AUC = 0.69) (Appendix A). We analyzed different tumor volume cutoff values (0.5 cc, 1.0 cc, 2.0 cc, 2.8 cc, 3.0 cc, 3.5 cc) and compared HR and *p*-values. The results confirmed that 2.8 cc is the optimal cut-off value as a predictive factor for BCR (Table 2). (The cutoff values of two tumor volumes with *p* < 0.0001 that were not selected (3.0 cc and 3.5 cc) were also verified by corresponding models, as shown in Appendix A).

Univariate and multivariate predictors for BCR obtained from cox proportional hazard analysis are shown in Table 2. The predictors for BCR were pathological stage T ≥ 3 (HR = 4.66 [95% CI: 2.81–7.73], *p* < 0.0001), positive surgical margin (HR = 4.18 [95% CI: 2.46–7.10], *p* < 0.0001), tumor volume ≥ 2.8 cc (HR = 3.10 [95% CI: 1.86–5.17], *p* < 0.0001), followed by PSA density ≥0.26 (HR = 2.06 [95% CI: 1.21–3.53], *p* = 0.0082), tumor located in the Posterior region (HR = 2.24 [95% CI: 1.07–4.65], *p* = 0.0314), tumor located in the PZ (HR = 3.28 [95% CI: 1.01–10.6], *p* = 0.0472). The multivariate analysis showed that the independent predictor of BCR was only tumor volume ≥ 2.8 cc (HR = 2.47 [95% CI: 1.14–5.36], *p* = 0.0225) (Table 2).

The Kaplan–Meier method was used to evaluate the PFS curve. The PFS of patients with tumors located in the PZ was inferior to those in the TZ (Figure 2A *p* = 0.0354). Furthermore, patients harboring tumors located in the posterior had shorter PFS than those in the anterior area (Figure 2B *p* = 0.027). Consistent with cox analysis, there was no significant difference between the PFS of the patients with tumors in the apex and not-apex area (Figure 2C *p* = 0.3135). PFS in the patients with tumor volume ≥ 2.8 cc was significantly inferior to those with less than 2.8 cc (Figure 2D *p* < 0.0001).

### 3.3. Model for Predicting PFS by Tumor Volume at Specific Location

Based on the analysis of clinical factors related to BCR in Table 2 and Figure 2, tumor volume and tumor location (PZ and Posterior location) were statistically significant predictive factors. Therefore, we established a risk classification model using tumor volume and location to stratify patients on the basis of risk of progression. The three risk factors that predict BCR in the model are tumor volume ≥ 2.8 cc, tumor located in PZ, and tumor located in the posterior area. The capability of the unfavorable risk to predict BCR was shown in Table 3 and only these risk factors predicted BCR on multivariable analysis (HR 3.16 [95% CI: 1.52–6.56], *p* = 0.002).

To further explore the predictive ability of the novel risk model, we divided the patients into the low-risk group, intermediate-risk group, and high-risk group according to the risk grouping of Pca by the American Cancer Society (ACS) [20] and validated the predictive value of the risk models among different ACS risk groups. In the analysis of the high-risk group, our unfavorable risk model could not predict disease progression independently (Table 4). However, the risk factors were the only independent predictor for PFS among patients with low to intermediate-risk groups (HR 4.43 [95% CI: 1.51–13.01], *p* = 0.0068) (Table 5).

### 3.4. Risk Model to Stratify Patient Prognosis

According to our established risk model, we divided the patients into three groups (favorable; displayed zero risk factors, moderate; displayed one or two risk factors, unfavorable; displayed all three risk factors). Overall, 61, 343, and 104 patients were classified as belonging to the favorable, moderate, and unfavorable group, respectively (Figure 3A).

The PFS curves of the three groups of patients (Figure 3B) showed that the PFS of the unfavorable group was significantly worse than that of the moderate group (*p* < 0.0001) and the favorable group (*p* = 0.001), while there was no significant difference between the moderate group and the favorable group (*p* = 0.1150).

The median tumor volume of the three groups was 1.33 cc, 1.81 cc, and 4.92 cc, respectively and there were significant differences between the three groups (Figure 3C).

In addition, we analyzed the impact of tumor volume on PFS in different prostate regions with the tumor volume of 2.8 cc as the threshold (Figure 4). The results suggested that the PFS of tumor ≥ 2.8 cc in the PZ is significantly worse than that of less than 2.8 cc (Figure 4A *p* < 0.0001). Similar results were observed for tumors ≥ 2.8 cc in the posterior location (Figure 4C *p* < 0.0001). Of note, the 2.8 cc cutoff value in TZ also showed a significant difference in PFS between the two groups (Figure 4B *p* = 0.0345). On the other hand, the significant difference was not seen in the anterior area (Figure 4D *p* = 0.0873).

## 4. Discussion

In our study, a tumor with a volume ≥ 2.8 cc was identified as an independent predictive factor for BCR (*p* = 0.0225). Furthermore, we established novel risk classification together with PZ and posterior location, which distinguished PFS between different risk groups. We believe this risk model will provide novel prognostic significance in patients who received RP.

Previous studies showed the positive surgical margin after RP is a potential predictive factor for BCR [21,22,23,24,25,26,27,28,29]. It is difficult to completely avoid the incidence of positive surgical margins through objective methods. Several studies found that positive surgical margins with limited length [30,31], locations [32], or quantity [33] decreased the correlation with BCR. Another study showed that tumor volume was associated with BCR in patients who underwent RP with negative surgical margins [34]. In addition, tumor volume and GS were even more significant predictors for BCR than positive margins [35] and the location of the tumor could predict the incidence of positive surgical margins [36,37,38,39]. Multivariate analysis showed that the predictive value of our risk model was superior to the positive surgical margin. These findings suggested that focusing on tumor volume and location, not only resection margins will give us better prognostic information in the treatment of localized Pca.

Regarding the prognostic significance of tumor localization, tumors originating in the TZ have been reported to be associated with a better prognosis in comparison with those in the PZ [39,40,41]. Augustin et al. found that the location of prostate cancer in the TZ was associated with better progression-free survival after RP (*p* = 0.0402) [40]. However, the zonal location offers no advantage over the well-established prognostic factors in predicting recurrence. Some more detailed anatomical differentiation (anterior, posterior, the apex of prostate, bladder neck) also revealed the difference in tumor location on prognosis [42,43]. Magheli et al. found that tumors in the anterior prostate were associated with favorable pathological features and improved biochemical-free survival, although it was not an independent predictor of BCR [42]. There are also some studies that have concluded that tumor location is not related to prognosis [44,45].

Tumor volume has been reported to show a significant correlation with BCR after RP [46,47,48,49,50]. Generally, tumor volume < 0.5 cc has been considered as an insignificant Pca, which has a low potential of recurrence [51]. The predictive factors for BCR in patients with low-volume prostate cancer (≤0.5 cc) have not been well studied [52]. Several reports proposed to increase the thresholds of volume for insignificant cancer to avoid over-treatment [14], however, other studies showed that the modified criteria had a higher risk of BCR in Gleason 4/5 cancer [53]. The tumor volume was superior to the percentage of cancer (tumor volume/prostate volume ratio) for predicting the prognosis after RP [54]. Different tumor volume cut-off values were proposed to determine the prognosis of Pca. Friedersdorff et al. suggested that tumor volume ≥ 5 cc (AUC = 0.79) was a significant prognostic factor for BCR [55]. Another study set the cut-off values as: minimal (≤1.0 cc), middle (1.1–5.0 cc), or extended (>5.0 cc) [47]. Shin et al. divided the tumor volume into three groups according to 2 cc and 5 cc, in multivariate analysis, recurrence-free survival could be independently predicted [56]. The tumor volume in the surgical specimen after neoadjuvant therapy was investigated and the study showed that patients with residual tumors ≥ 1.0 cc in the specimen had a higher risk of BCR [57]. Raison et al. studied 685 British patients who underwent laparoscopic and robot-assisted RP and revealed that 2.5 cc (AUC = 0.71) was the best cutoff value for predicting BCR [58]. Of note, some studies showed that the tumor volume alone may not be able to evaluate the prognosis of recurrence and prognosis after RP [13,59]. O’Neil et al. suggested that tumors in some locations are larger and more likely to invade the sites that are prone to recurrence [37]. However, there have been no studies that have analyzed the prognostic value of tumor volume combined with tumor localization.

In our study, we attempted to evaluate the potential interaction between tumor volume and location, the tumor volume cutoff value obtained by the ROC curve was 2.8 cc (AUC = 0.69). Therefore, we used the tumor volume threshold (≥2.8 cc) of the specific location to improve the capability of our risk model. We hypothesized that the larger tumor volume in the PZ and/or posterior of the prostate may be associated with BCR. Our findings demonstrated that the prognostic significance of tumor volume over 2.8 cc varied by tumor localization (Figure 4). In our model, the interaction between prostate tumor location and volume was a promising predictor of prostate BCR. Interestingly, our risk model was an independent predictor in patients with low and intermediate risk while it was not in patients with high risk. Extended dissection during surgery and close follow-up after surgery may enhance clinical benefit in patients who met our criteria.

The limitations of this study are as follows. First, our study included a single Asian race. Compared with the western population, the Asian population has a lower incidence and mortality of prostate cancer [60]. The tumor volume of African American men with prostate cancer is larger than that of white men [61]. The risk of BCR in black Americans has been reported to be 1.6 times higher than that in white Americans [62]. These results suggested that there may be differences in clinical and pathological features between races. Further validation of our risk model will be warranted in other patients’ cohorts. Second, our study may need to be further investigated using genomic analysis. The previous study has revealed that prostate cancer risk alleles are associated with prostate cancer volume and prostate size [63]. Downregulation of PAH and AOC1 and upregulation of DDC, LIN01436, and ORM1 were associated with the development of prostate cancer [8,64]. Molecular and cellular biological studies are also closely related to the site of prostate tumorigenesis [41]. Studying the specific genes behind it could improve understanding of the region or cell-type characteristics of prostate cancer. These features account for differences in tumor progression and invasion between different regions of the prostate [41]. The unique biological characteristics of tumor types in different prostate regions can help guide individualized treatment and patient risk stratification. Finally, further validation of our clinical parameters using the latest imaging system PSMA/PET [65] or artificial intelligence system (deep learning) [66] may enhance the clinical importance of this study.

## 5. Conclusions

Tumor volume ≥ 2.8 cc was an independent predictive factor for BCR in patients who received RP. Furthermore, we established a novel risk model using tumor volume over 2.8 cc and tumor location (PZ and/or posterior). Our risk classification could predict patient prognosis and will help us to optimize peri-operative and post-operative treatment strategies.

## Figures and Tables

**Figure 1 cancers-14-05823-f001:**
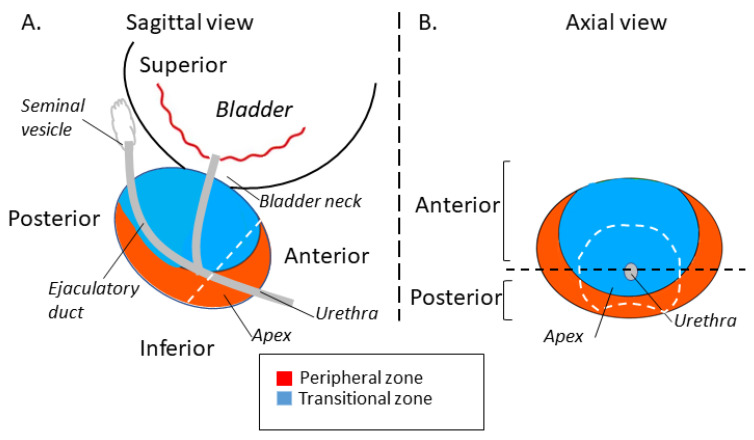
Schematic diagram of an anatomical division of the prostate. The location of the Anterior/Posterior and Peripheral/Transitional Zones are described. (**A**) Sagittal view. (**B**) Axial view.

**Figure 2 cancers-14-05823-f002:**
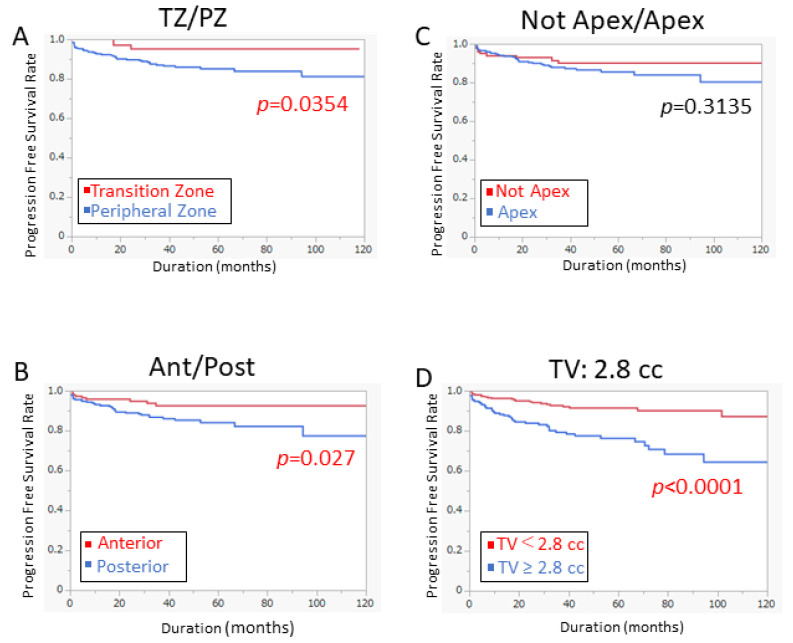
Prognostic significance of tumor location and tumor volume. (**A**) Patients with tumor in the PZ had significantly worse PFS than those in the TZ (*p* = 0.0354). (**B**) Patients with tumor in the posterior region had significantly worse PFS than those in the anterior region (*p* = 0.027). (**C**) There was no difference in PFS between apex and non-apex regions. (**D**) Patients with tumor volume ≥ 2.8 cc had significantly worse PFS than those <2.8 cc (*p* < 0.0001).

**Figure 3 cancers-14-05823-f003:**
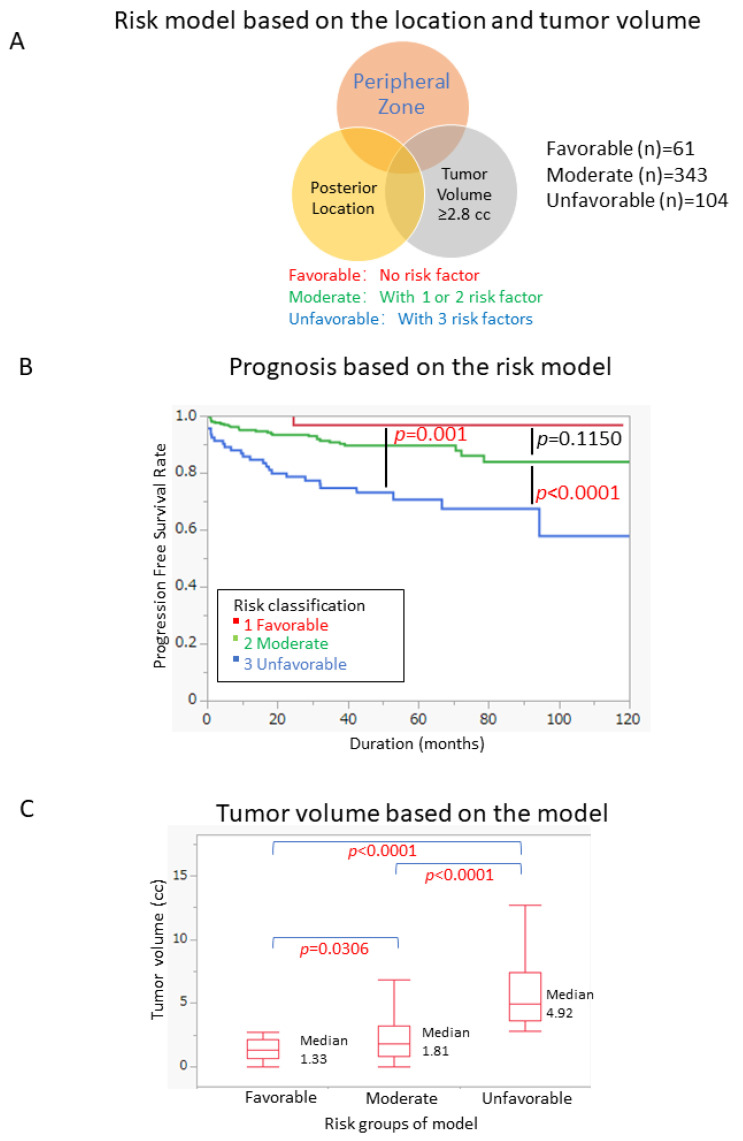
Prognostic model based on the location and tumor volume (**A**) Venn diagram of risk model based on the location and tumor volume. (**B**) Risk classification significantly differentiated the PFS between the Favorable and Unfavorable group (*p* = 0.001) and the Moderate and Unfavorable group (*p* < 0.0001). (**C**) The tumor volume showed significant differences among different risk groups.

**Figure 4 cancers-14-05823-f004:**
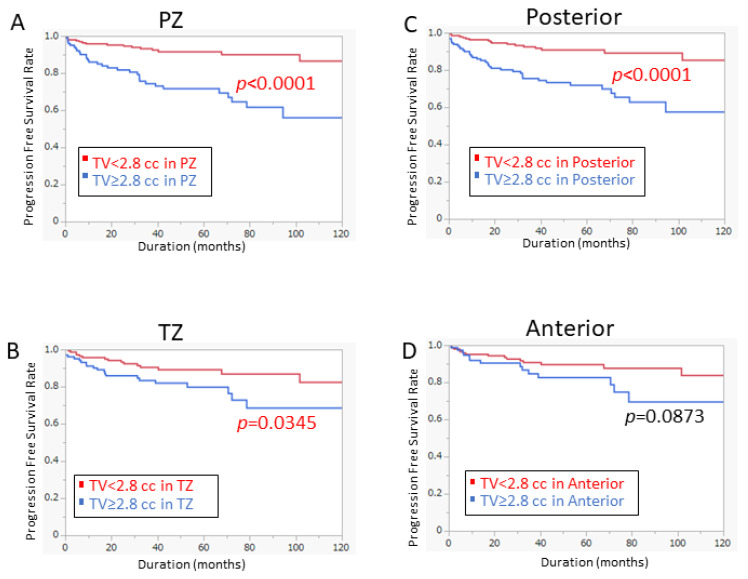
Prognostic significance of Tumor volume 2.8 cc based on the location. (**A**) Patients with tumor volume ≥ 2.8 cc had significantly worse PFS in the PZ (*p* < 0.0001). (**B**) Patients with tumor volume ≥ 2.8 cc had significantly worse PFS in the TZ (*p* = 0.0345). (**C**) Patients with tumor volume ≥ 2.8 cc had significantly worse PFS in the posterior region (*p* < 0.0001). (**D**) In the anterior region, there was no difference in PFS by tumor volume cutoff of 2.8 cc.

**Table 1 cancers-14-05823-t001:** Characteristics of patients.

Characteristics	
Number of patients	557
Median age at operation (range), years	67 (46–77)
Median follow-up time (range), months	45.3 (12–161.5)
Median initial PSA (range) (ng/mL)	7.71 (2.15–87.16)
Gleason score sum, n (%)	
≤7	444 (79.7)
8	48 (8.6)
≥9	61 (11.0)
T stage, n (%)	
≤2b	195 (35.0)
≥2c	361 (64.8)
Risk Group; Low/Intermediate/High, n (%)	77 (13.8)/279 (50.1)/201 (36.1)
Tumor Volume (range), cc	2.12 (0.02–57)
Tumor Location, n (%)	
apex	355 (63.7)
middle	353 (63.4)
bladder neck	119 (21.4)
Tumor Location, n (%)	
anterior	268 (48.1)
posterior	292 (52.4)
Tumor Location, n (%)	
PZ	374 (67.1)
TZ	208 (37.3)
N stage, n (%)	
positive	8 (1.4)
Seminal Vesicle Invasion, n, (%)	48 (8.6)
Extracapsular Extension, n, (%)	138 (24.8)
Resection Margins, n, (%)	169 (30.3)
PSA Recurrence, n, (%)	66 (11.8)

PSA = prostate-specific antigen; T stage = tumor stage; N stage = lymph node stage; PZ = peripheral zone; TZ = transition zone.

**Table 2 cancers-14-05823-t002:** Univariable and multivariable cox proportional hazard regression models in predictive factors for PFS in localized Pca (overall risk).

	Univariable				Multivariable		
	Cut Off	HR	95% CI	*p* Value	HR	95% CI	*p* Value
Age	≥67	0.96	0.59–1.57	0.8842			
initial PSA	≥7.71 ng/mL	1.65	1.00–2.73	0.0505			
PSAD	≥0.26	2.06	1.21–3.53	0.0082	1.51	0.73–3.09	0.2643
GS	≥7	1.15	0.46–2.88	0.7593			
T stage	≥T3	4.66	2.81–7.73	<0.0001	1.69	0.77–3.71	0.1894
RM	positive	4.18	2.46–7.10	<0.0001	1.99	0.94–4.20	0.0712
Tumor location							
	Apex	1.45	0.70–3.02	0.3166			
	PZ	3.28	1.01–10.60	0.0472	2.21	0.49–10.05	0.3030
	posterior	2.24	1.07–4.65	0.0314	1.72	0.72–4.12	0.2193
TV							
	≥0.5 cc	1.61	0.73–3.53	0.2344			
	≥1.0 cc	2.18	1.11–4.27	0.0240			
	≥2.0 cc	2.74	1.55–4.82	0.0005			
	≥2.8 cc **	3.10	1.86–5.17	<0.0001	2.47	1.14–5.36	0.0225 *
	≥3.0 cc	2.96	1.80–4.88	<0.0001			
	≥3.5 cc	2.80	1.72–4.58	<0.0001			

PSA = prostate-specific antigen; PSAD = prostate-specific antigen density; GS = Gleason score; T stage = tumor stage; RM = resection margins; HR = hazard ratio; CI = confidence interval; * *p*-value < 0.05, ** tumor volume cutoff value based on the ROC curve.

**Table 3 cancers-14-05823-t003:** Univariable and multivariable cox proportional hazard regression models in predictive factors for PFS in localized Pca (overall risk) with unfavorable risk.

	Univariable				Multivariable		
	Cut Off	HR	95% CI	*p* Value	HR	95% CI	*p* Value
Age	≥67	0.96	0.59–1.57	0.8842	-	-	-
initial PSA	≥7.71 ng/mL	1.65	1.00–2.73	0.0505	-	-	-
PSAD	≥0.26	2.06	1.21–3.53	0.0082	1.55	0.76–3.15	0.2307
GS	≥7	1.15	0.46–2.88	0.7593	-	-	-
T stage	≥T3	4.66	2.81–7.73	<0.0001	1.64	0.74–3.65	0.2261
RM	positive	4.18	2.46–7.10	<0.0001	2.09	0.99–4.42	0.0548
Unfavorable Risk	PZ + Post + TV2.8 cc	4.74	2.60–8.65	<0.0001	3.16	1.52–6.56	0.0020 *

PSA = prostate-specific antigen; PSAD = prostate-specific antigen density; GS = Gleason score; T stage = tumor stage; RM = resection margins; PZ + Post + TV2.8 cc = tumor volume ≥ 2.8 cc in posterior location of peripheral zone; HR = hazard ratio; CI = confidence interval; * *p*-value < 0.05.

**Table 4 cancers-14-05823-t004:** Univariable and multivariable cox proportional hazard regression models in predictive factors for PFS in localized Pca (high risk).

	Univariable				Multivariable		
	Cut Off	HR	95% CI	*p* Value	HR	95% CI	*p* Value
Age	≥67	0.76	0.40–1.47	0.4167	-	-	-
initial PSA	≥7.71 ng/mL	1.04	0.52–2.08	0.9097	-	-	-
PSAD	≥0.26	1.9	0.82–4.40	0.1326	-	-	-
GS	≥7	1.29	0.18–9.46	0.7991	-	-	-
T stage	≥T3	4.38	2.11–9.10	<0.0001	1.98	0.75–5.25	0.1701
RM	positive	4.65	2.16–10.02	<0.0001	2.37	0.95–5.91	0.0649
Unfavorable Risk	PZ + Post + TV2.8 cc	3.5	1.64–7.47	0.0012	1.87	0.77–4.53	0.1653

PSA = Prostate Specific Antigen; PSAD = Prostate Specific Antigen Density; GS = Gleason Score; T stage = Tumor Stage; RM = Resection Margins; HR = Hazard Ratio; CI = Confidence Interval.

**Table 5 cancers-14-05823-t005:** Univariable and multivariable cox proportional hazard regression models in predictive factors for PFS in localized Pca (low to intermediate risk).

	Univariable				Multivariable		
	Cut Off	HR	95% CI	*p* Value	HR	95% CI	*p* Value
Age	≥67	1.07	0.51–2.25	0.8546	-	-	-
initial PSA	≥7.71 ng/mL	1.56	0.74–3.28	0.2458	-	-	-
PSAD	≥0.26	1.52	0.72–3.19	0.2716	-	-	-
GS	≥7	0.74	0.26–2.15	0.5855	-	-	-
T stage	≥T3	3.34	1.59–7.01	0.0015	0.97	0.28–3.38	0.961
RM	positive	3.03	1.42–6.47	0.0043	1.38	0.43–4.41	0.5904
Unfavorable Risk	PZ + Post + TV2.8 cc	4.71	1.75–12.69	0.0022	4.43	1.51–13.01	0.0068 *

PSA = prostate-specific antigen; PSAD = prostate-specific antigen density; GS = Gleason score; T stage = tumor stage; RM = resection margins; HR = hazard ratio; CI = confidence interval; PZ + Post + TV2.8 cc = tumor volume ≥ 2.8 cc in posterior location of the peripheral zone. * *p*-value < 0.05.

## Data Availability

The data presented in this study are available on request from the corresponding author.

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
