# Peer review of "Tumor Location and a Tumor Volume over 2.8 cc Predict the Prognosis for Japanese Localized Prostate Cancer"

_cancers, 2022, doi:10.3390/cancers14235823_

Round 1
Reviewer 1 Report
It’s a retrospective study of 557 patients with prostate cancer who underwent radical prostatectomy. The goal of this study was to evaluate if the tumor volume plays a role as a prognostic factor in the predictions and estimation of the recurrence of prostate cancer after radical prostatectomy. Ver a well-written paper. I congratulate the authors for their work. I have some points to be considered and discussed by the authors in their manuscript.
1) Why a short follow-up? The authors should explain that in their manuscript since most patients develop a relapse within the first two years. It’s also interesting to read if there is another prediction factor after five years of follow-up after surgery and to see the PSA biochemical relapse within the first five after in high-risk patients who underwent radical prostatectomy.
2) In Table 1: since cT2b is the border between intermediate risk and high risk because cT2c is a high risk, I prefer to make the table between high risk and intermediate risk. So, I advise the authors to also change the T stage from ≤ cT2b and ≥cT2c so that we can know the number of patients with intermediate-risk at the operation time.
Author Response
Reviewer #1
It’s a retrospective study of 557 patients with prostate cancer who underwent radical prostatectomy. The goal of this study was to evaluate if the tumor volume plays a role as a prognostic factor in the predictions and estimation of the recurrence of prostate cancer after radical prostatectomy. Ver a well-written paper. I congratulate the authors for their work. I have some points to be considered and discussed by the authors in their manuscript.
- Why a short follow-up? The authors should explain that in their manuscript since most patients develop a relapse within the first two years. It’s also interesting to read if there is another prediction factor after five years of follow-up after surgery and to see the PSA biochemical relapse within the first five after in high-risk patients who underwent radical prostatectomy.
A: We thank the Reviewer #1 for careful reviewing and kindly providing very helpful comments to improve our manuscript.
According to the suggestions of the reviewers, we added the follow-up time. Follow-up terms ranged from 12 to 161.5 months, with a median follow-up time of 45.3 months. (Page4 Line 129-130)
We have also added the relevant information into Table 1. (Page 4 Table1)
We also reanalyzed the data of patients with biochemical recurrence as suggested. They were divided into those who had BCR within five years and those who had BCR after five years later. Patient backgrounds of the two groups were compared in the following table. No significant differences were observed between two groups.
|
All BCR patients=66 |
BCR within 5 years |
BCR after 5 years |
p value |
|
Number of patients (%) |
57 (86.4%) |
9 (13.6%) |
|
|
Median initial PSA (range) (ng/ml) |
10.76 (3.63-47.35) |
7.66 (4.29-12.52) |
0.0844 |
|
Gleason score sum, n |
|
||
|
≤ 7 |
29 |
6 |
|
|
≥ 8 |
28 |
3 |
0.3480 |
|
T stage, n |
|
||
|
≤ 2b |
12 |
2 |
|
|
≥ 2c |
45 |
7 |
0.9367 |
|
Tumor Volume (range), ml |
3.6 (0-57) |
3.396 (0.192-7.548) |
0.4378 |
|
Tumor Location, n |
|
||
|
anterior |
9 |
0 |
0.2381 |
|
posterior |
33 |
3 |
|
|
Tumor Location, n |
|
||
|
PZ |
40 |
3 |
0.5175 |
|
TZ |
3 |
0 |
|
|
N stage, n (%) |
|
||
|
positive |
4 (7%) |
0 |
0.2966 |
|
Seminal Vesicle Invasion, n |
20 |
1 |
0.1127 |
|
Extracapsular Extension, n |
30 |
7 |
0.3269 |
|
Resection Margins, n |
35 |
7 |
0.5420 |
2) In Table 1: since cT2b is the border between intermediate risk and high risk because cT2c is a high risk, I prefer to make the table between high risk and intermediate risk. So, I advise the authors to also change the T stage from ≤ cT2b and ≥cT2c so that we can know the number of patients with intermediate-risk at the operation time.
A: We thank the Reviewer #1 for careful reviewing and kindly providing very helpful comments to improve our manuscript.
According to the reviewer's comment, we re-edited the data of T stage in Table 1. We also specify the number and proportion of patients in each risk group. There were 195 (35.0%) patients with T stage ≤2b and 361 (64.8%) patients with T stage ≥2c. There were 77 (13.8%) patients in the low-risk group, 279 (50.1%) patients in the intermediate-risk group, and 201 (36.1%) patients in the high-risk group. We have also modified the corresponding description in lines 133-137 of page 4. (Page4 Line 133-137)

Reviewer 2 Report
Thank you for inviting me to review the article titled “The Location of Tumor Volume Over 2.8cc Predict the Prognosis Among Japanese Localized Prostate Cancer”
In this Manuscript, authors develop a new model based on tumor volume and location to predict the risk of biochemical risk of recurrence.
The paper is written eloquently, and the results are presented appropriately and the discussion is to the point to address the finding.
It seems like the tumor volume is probably driving the model with the posterior location, with a possibly smaller impact of the peripheral zone location.
Criticisms/comments:
Minor:
· The number of volume 2.8 cc with an AUC of 0.69 is a bit soft choice, and a higher volume might have better true positive values (higher AUC), it would be interesting to see if the model performs the same for higher volume cut off eg. 3 or 3.5cc.
· Line 62-63 – needs newer and more than one reference.
Author Response
Reviewer #2
Thank you for inviting me to review the article titled “The Location of Tumor Volume Over 2.8cc Predict the Prognosis Among Japanese Localized Prostate Cancer”
In this Manuscript, authors develop a new model based on tumor volume and location to predict the risk of biochemical risk of recurrence.
The paper is written eloquently, and the results are presented appropriately and the discussion is to the point to address the finding.
It seems like the tumor volume is probably driving the model with the posterior location, with a possibly smaller impact of the peripheral zone location.
Criticisms/comments:
Minor:
- The number of volume 2.8 cc with an AUC of 0.69 is a bit soft choice, and a higher volume might have better true positive values (higher AUC), it would be interesting to see if the model performs the same for higher volume cut off eg. 3 or 3.5cc.
A: We thank the Reviewer #2 for careful reviewing and kindly providing very helpful comments to improve our manuscript.
Just as the reviewer said, the 3.0cc & 3.5cc tumor volume cutoff value can also be distinguished and predicted in the model. Moreover, the differences of PFS between the Favorable group and the Unfavorable group of these two models (3.0cc p=0.0008; 3.5cc p=0.0001) are more significant than the existing model (p=0.001).
However, we performed a regression analysis for different tumor volume cutoff values in the study (Table2). A comparison of different HRs (Hazard ratios) shows that the cutoff value of 2.8cc has the largest HR value when p < 0.0001. Considering the result of HR regression analysis and the possibility of false positive prediction of BCR brought by larger tumor cutoff value, we would prefer the cutoff value of 2.8cc.
We have added the results of regression analysis of TV≥3.5cc in Table 2. (Page5 Table2)
We submit the unselected 3.0cc and 3.5cc models as supplementary figures. (Page5 Line 162-164, and Page12 Line 342-348)
(Supplementary figure 2 and 3 can be viewed in the word file)  
|
Univariable |
|||
|
TV Cut off |
HR |
95% CI |
P value |
|
≥0.5cc |
1.61 |
0.73-3.53 |
0.2344 |
|
≥1.0cc |
2.18 |
1.11-4.27 |
0.024 |
|
≥1.5cc |
2.52 |
1.37-4.63 |
0.003 |
|
≥2.0cc |
2.74 |
1.55-4.82 |
0.0005 |
|
≥2.5cc |
2.71 |
1.62-4.54 |
0.0002 |
|
≥2.8cc |
3.1 |
1.86-5.17 |
<0.0001 |
|
≥3.0cc |
2.96 |
1.80-4.88 |
<0.0001 |
|
≥3.5cc |
2.80 |
1.72-4.58 |
<0.0001 |
- Line 62-63 – needs newer and more than one reference.
A: We thank the Reviewer #2 for careful reviewing and kindly providing very helpful comments to improve our manuscript.
As suggested by the reviewer, we added references supporting this view of “Recently, several studies proposed the novel definition of insignificant prostate cancer as a tumor volume of less than 2.5cc” in lines 62-63 of page 2. Among the references [13-17], two were the most recent 2019-2020 references with 2.5cc cutoff-value. We also added another research reference in 2019 that took 2.0cc as the cutoff-value [18].
We sorted and adjusted the corresponding references.
- Ito, Y.; Udo, K.; Vertosick, E.A.; Sjoberg, D.D.; Vickers, A.J.; Al-Ahmadie, H.A.; Chen, Y.B.; Gopalan, A.; Sirintrapun, S.J.; Tickoo, S.K.; et al. Clinical Usefulness of Prostate and Tumor Volume Related Parameters following Radical Prostatectomy for Localized Prostate Cancer. J Urol 2019, 201, 535-540, doi:10.1016/j.juro.2018.09.060.
- Ting, F.; van Leeuwen, P.J.; Delprado, W.; Haynes, A.M.; Brenner, P.; Stricker, P.D. Tumor volume in insignificant prostate cancer: Increasing the threshold is a safe approach to reduce over-treatment. Prostate 2015, 75, 1768-1773, doi:10.1002/pros.23062.
- Fugini, A.V.; Antonelli, A.; Giovanessi, L.; Gardini, V.C.; Abuhilal, M.; Zambolin, T.; Tardanico, R.; Simeone, C.; Cunico, S.C. Insignificant Prostate Cancer: Charateristics and Predictive Factors. Urologia Journal 2011, 78, 184-186, doi:10.5301/ru.2011.8541.
- Antonelli, A.; Vismara Fugini, A.; Tardanico, R.; Giovanessi, L.; Zambolin, T.; Simeone, C. The percentage of core involved by cancer is the best predictor of insignificant prostate cancer, according to an updated definition (tumor volume up to 2.5 cm3): analysis of a cohort of 210 consecutive patients with low-risk disease. Urology 2014, 83, 28-32, doi:10.1016/j.urology.2013.07.056.
- Yamada, Y.; Sakamoto, S.; Sazuka, T.; Goto, Y.; Kawamura, K.; Imamoto, T.; Nihei, N.; Suzuki, H.; Akakura, K.; Ichikawa, T. Validation of active surveillance criteria for pathologically insignificant prostate cancer in Asian men. Int J Urol 2016, 23, 49-54, doi:10.1111/iju.12952.
- Frankcombe, D.E.; Li, J.; Cohen, R.J. Redefining the Concept of Clinically Insignificant Prostate Cancer. Urology 2020, 136, 176-179, doi:10.1016/j.urology.2019.10.019.
